# Development of an Add-On Device Using 3D Printing for the Enhancement of Drug Administration Efficiency of Dry Powder Inhalers (Accuhaler)

**DOI:** 10.3390/pharmaceutics14091922

**Published:** 2022-09-12

**Authors:** Kittipat Suwanpitak, Lee-Yong Lim, Inderbir Singh, Pornsak Sriamornsak, Thanongsak Thepsonthi, Kampanart Huanbutta, Tanikan Sangnim

**Affiliations:** 1Faculty of Pharmaceutical Sciences, Burapha University, 169, Seansook, Muang, Chonburi 20131, Thailand; 2Division of Pharmacy, School of Allied Health, University of Western Australia, Perth, WA 6009, Australia; 3Chitkara College of Pharmacy, Chitkara University, Patiala 140401, Punjab, India; 4Faculty of Pharmacy, Silpakorn University, Nakhon Pathom 73000, Thailand; 5Faculty of Engineering, Burapha University, 169, Seansook, Muang, Chonburi 20131, Thailand; 6School of Pharmacy, Eastern Asia University, Thanyaburi, Pathumthani 12110, Thailand

**Keywords:** Accuhaler, 3D printing technology, COPD, dry powder inhalers, computational fluid dynamics

## Abstract

The goal of this study was to develop an add-on device for dry powder inhalers (Accuhaler) via 3D printing to improve drug administration efficiency in patients with limited inspiratory capacity, including young children, the elderly, and those with chronic obstructive pulmonary disease. With salmeterol xinafoate and fluticasone propionate as model active pharmaceutical ingredients (API), the emitted API doses were used to assess the effectiveness of the add-on device. The APIs were quantified by an HPLC assay validated for specificity, range, linearity, accuracy, and precision. The motor power of the add-on device could be regulated to moderate fan speed and the air flow in the assembled device. When 50–100% of the fan motor power of the add-on device was used, the emitted dose from the attached dry powder inhaler (DPI) was increased. A computational fluid dynamics application was used to simulate the air and particle flow in the DPI with the add-on device in order to elucidate the operating mechanism. The use of the add-on device combined with a sufficient inhalation flow rate resulted in a larger pressure drop and airflow velocity at the blister pocket. As these characteristics are associated with powder fluidization, entrainment, and particle re-suspension, this innovative add-on device might be utilized to enhance the DPI emitted drug dose for patients with low inspiratory rates and to facilitate the provision of adequate drug doses to achieve the treatment outcomes.

## 1. Introduction

Lung disorders such as asthma and chronic obstructive pulmonary disease are treated mostly with inhaled medications [1] however, the desired treatment outcomes require the inhalation devices to be used correctly [2,3,4]. Inhalation devices can be broadly classified into metered dose inhalers (MDI) or dry powder inhalers (DPI). MDI contains propellants to expel the drug from the device whereas optimal drug delivery using a DPI is dependent on the patient providing adequate inspiratory flow rate/force and the drug powder having the desired particle sizes [5]. DPI requires a minimum inspiratory flow rate of 30 L/min for efficient drug delivery [6]. However, a study by Al-Showair et al. has found older patients (average age 72.5 years) to have an impaired forced expiratory volume (FEV) of less than 47.8% and an average inspiratory flow rate of less than 30 L/min that then resulted in a reduced mean DPI emitted dose of only 57% [7].

A patient’s inspiratory flow rate affects not only the amount of drug delivered from a DPI, but also the shape and size distribution of the emitted drug particles which in turn influence the DPI powder flow and deposition along the patient’s respiratory tract [8]. Compared to large particles, smaller particles have a less hindered flow in the respiratory tract and therefore a higher potential to reach the alveolar region in the deep lungs. This flow pattern together with the larger specific surface area that enables smaller particles to also dissolve faster in vivo ultimately leads to improved drug bioavailability from the DPI. This has been demonstrated by Haidl and colleagues, who showed that an increase in inspiratory flow rate in the range of 10–60 L/min promoted drug particle attrition, and the resultant smaller drug particles did not only increase the emitted dose, but also improved the drug dissolution rate and the delivery of a higher drug dose in the alveoli [9]. Proper DPI techniques are therefore essential to provide efficient drug delivery and optimal therapeutic outcomes.

Not all patients are able to generate the threshold inspiratory flow rate/force required for effective DPI medication administration. Vulnerable patients include young children, the elderly, and patients suffering from severe asthma and COPD who ironically are most in need of efficient DPI drug delivery. We hypothesize that an add-on device may be developed for use in conjunction with a DPI to mitigate a patient’s suboptimal inspiratory rates so as to deliver in a consistent manner the target drug dose to the patient. Therefore, the idea of designing an accessory to improve airflow in a DPI was conceived.

Nowadays, preparing the conceptual add-on device is possible using 3D printing technology. The 3D printing technique increasingly gains interest in the field of pharmaceutical and medical research and application. This is because this technique shows the feasibility to promptly prepare the complex structure of pharmaceutical dosage forms and medical devices which can obtain maximum therapeutic outcomes with low side effects [10,11,12,13]. This technology was invented for the first time in the 1980s, although it has only recently gained widespread use [14]. In the printing process, numerous 3D fabrication techniques have been developed. Among these are stereolithography, digital light processing, fused deposition modeling, selective laser sintering, and binder jetting [15]. Each technique is compatible with a variety of printing materials and produces goods with distinct qualities. Due to its ease of use, affordability, and availability of printing materials, fused deposition modeling (FDM) for 3D printing has been actively utilized in pharmaceutical research [16,17]. As a consequence, FDM 3D printing was applied to fabricate the concept of an add-on device for DPI in this work.

The aim of this study was to use 3D printing to develop the add-on device for the enhancement of airflow in a DPI. The Accuhaler device containing drug combination of salmeterol xinafoate and fluticasone propionate was chosen as the representative DPI since it is the most often prescribed DPI for the treatment of asthma and COPD. The add-on device was fabricated by the fused deposition modeling (FDM) 3D-printing technique, a promising technology for prototyping equipment as it is a rapid, convenient, and affordable method and manufactures robust products [12]. The prototype add-on device was designed to contain a centrifugal fan, which when used with the Accuhaler, aims at facilitating the flow of drug particles and particle attrition to generate smaller sized particles in the DPI. The centrifugal fan’s flow rate could be regulated to meet the dosing needs of patients with varied inspiratory forces. In this study, we report on the effects of the centrifugal fan speed, airflow rate, and emitted dose uniformity of the modified DPI. In addition, the computational fluid dynamics of drug particles in the DPI were generated, and a two-factor-central composite design was applied to evaluate the relationship and effects of the inhalation flow rate and the device input power on the emitted doses of salmeterol xinafoate and fluticasone propionate. The two drugs were quantified by validated high performance liquid chromatography (HPLC) assays. Finally, the particle least square model was generated to confirm that the 3D printed prototype add-on device may be applied to a variety of patient groups.

## 2. Materials and Methods

### 2.1. Materials

Accuhaler, a commercial dry powder inhaler, was purchased from GlaxoSmithKline (Brentford, UK). Salmeterol xinafoate RS Batch No. SX/WS/001/20 was purchased from Vamsi Labs Ltd. (Maharashtra, India). Fluticasone propionate RS Batch No. NZ2513M was purchased from NewChem Co., Ltd. (Verona, Italy). An acrylonitrile–butadiene–styrene (ABS) filament was obtained from Beijing Tiertime Technology Co., Ltd. (Beijing, China). Acetonitrile was purchased from RCI Labscan Co., Ltd. (Bangkok, Thailand). All other chemicals and reagents were of analytical grade.

### 2.2. Method

#### 2.2.1. DPI Device Design

The DPI add-on device model (Figure 1a) was designed by the Shapr3D program in the format of a CAD (Computer-Aided Design) file that was compatible with the 3D printing process. The device was intended to be attached to the Accuhaler’s cover as illustrated in Figure 1b and the design model was based on the centrifugal or radial fan concept to generate airflow to transport drug particles throughout the device. The fan was designed in the form of a backward curve, which offers a more stable air flow with a low motor workload compared to the forward-curved fan [18]. A power button was included, which upon activation would generate the air to cause the drug particles to flow through the air channel. Figure 2 depicts the electrical circuit of the DPI add-on device. The motor is controlled by a pulse width modulation (PWM) speed regulation and is powered by a 12 V lithium battery.

#### 2.2.2. 3D Printing of DPI Add-On Device

The DPI add-on equipment model was printed by 3D printing (UP mini 2, Tiertime, California, USA). The CAD file format for the model design was sliced using the Upstudio software, a 3D printer software, into an STL file format. The acrylonitrile–butadiene–styrene (ABS) filament was employed as the printing material and the operational parameters in Table 1 were applied for the printing process.

#### 2.2.3. Evaluation of DPI Add-On Device

Evaluation of the DPI add-on equipment was performed under conditions that simulated the use of the equipment in a practical clinical setting and is described in the following sections. The add-on device was attached to the Accuhaler containing salmeterol xinafoate and fluticasone propionate.

##### Fan Speed and Airflow Rate

The fan speed in the add-on equipment was monitored using a digital tachometer (DT-2234C+, True Sense Technologies, India) and reported in units of RPM (rounds per minute). The input power was varied, and the fan response speed was recorded. The corresponding airflow rate in the Accuhaler assembled with the add-on equipment was also measured in units of L/min using the anemometer (BENETECH GM816, Shenzhen Jumaoyuan Science and Technology Co., Ltd., Shenzhen, China).

##### Emitted Dose Uniformity

The emitted dose uniformity is a method for evaluating the uniformity of each administered dose of medication. This is a critical evaluation subject since it determines the amount of medication a patient can take with every inhalation. Because the developed add-on equipment affects the flow of medication particles, it is essential to certify that all or the vast majority of drugs were delivered. In the test, the amounts of drugs (salmeterol xinafoate and fluticasone propionate) emitted from the DPI alone, and from the DPI assembled with the add-on equipment, were evaluated using the USP Apparatus B sampling apparatus for inhalation powders(6). To conduct the test, the DPI was connected to the USP Apparatus B and the vacuum was applied for 2 s, with each vacuum rate reflecting a specified inspiratory flow rate (10–60 L/min), and the fan power of the add-on equipment was increased from 0 to 100 percent to generate air flow. The sample collection chamber and filter were removed and rinsed thrice with acetonitrile:phosphate buffer (70:30), and the rinses were combined and sonicated to ensure complete drug dissolution. The final solution was filtered through a 0.45 µm nylon filter into an amber-colored vial and the amounts of salmeterol xinafoate and fluticasone propionate in the solution were quantified by HPLC assay. The emitted dose uniformity test was conducted three times.

##### Method Validation of HPLC Assay

Salmeterol xinafoate and fluticasone propionate in the experimental samples were simultaneously quantified according to the assay monographs published in the United States Pharmacopoeia [19] and using an HPLC with a photodiode array detector (HPLC-PDA detector) (DGU-20A5R, Shimadzu, Kyoto, Japan. The applied HPLC conditions are listed in Table 2, and the assay method was validated for specificity, range, linearity, accuracy, and precision [20,21].

##### Specificity

Specificity is defined as the ability of the assay to quantify salmeterol xinafoate in the presence of fluticasone propionate, and vice versa. Specificity was evaluated by adequate peak separation measured through the specific retention times obtained for salmeterol xinafoate and fluticasone propionate in the HPLC chromatogram. A standard solution salmeterol xinafoate RS and standard solution fluticasone propionate RS were dissolved in acetonitrile:phosphate buffer (70:30 *v*/*v*) to obtain a concentration of 0.75 µg/mL and 2.5 µg/mL, respectively. Then the combined solution was injected in the HPLC for analysis.

##### Range and Linearity

The range and linearity of the calibration curves for the two reference drugs were determined as follows. Standard solutions were prepared by dissolving salmeterol xinafoate RS and fluticasone propionate RS in acetonitrile:phosphate buffer (70:30 *v*/*v*) at concentrations of 0.1875–1.125 µg/mL and 1.25–3.75 µg/mL, respectively. The standards were analyzed in the HPLC, and the AUC of the drug peak was recorded and plotted against the respective drug concentration to calculate the correlation coefficient (R^2^).

##### Accuracy

Assay accuracy was evaluated by spiking a test solution (salmeterol xinafoate RS and fluticasone propionate RS) with known concentrations then analyzing the spiked solutions using the HPLC assay. Accuracy as measured by percent recovery was determined using Equation (1). Acceptable accuracy as defined by the AOAC 2019 standards is 80–110% recovery [22].
(1)$$\mathrm{Recovery}\;\%=\frac{\mathrm{Theoretical}\;\mathrm{concentration}\;\mathrm{spiked}-\mathrm{Actual}\;\mathrm{determined}\;\mathrm{concentration}\;}{\mathrm{Theoretical}\;\mathrm{concentration}\;\mathrm{spiked}}\times 100$$

##### Precision

Salmeterol xinafoate RS and fluticasone propionate RS were mixed at concentrations of 0.75 g/mL and 2.5 g/mL, respectively, to evaluate the assay precision. The standard solution was analyzed six times and the percent relative standard deviation (RSD) was determined. RSD less than 7.3% was deemed acceptable according to the AOAC 2019 guideline.

#### 2.2.4. Computation Fluid Dynamics

To understand air flow mechanism and direction, the air and particle flow from the DPI were simulated using the computational fluid dynamics program (SOLIDWORKS Flow Simulation). Computer-aided design (CAD) modelling of the Accuhaler with or without the add-on equipment was created, as presented in Figure 3 and the flow field, pressure drop, and flow velocity were monitored during the simulation. Boundary conditions were defined by setting the mouthpiece outlet volume flow at three levels, 10, 30, and 60 L/min, and setting the environment pressure (131 and 325 Pa) at the by-pass channel and the air inlet (Accuhaler air inlet or add-on device fan inlet). The boundary condition of the by-pass channel was defined by the environmental pressure as the channel is an internal opening that is in contact with the environmental pressure. The fan rotating speed for the add-on equipment was set based on data obtained in the fan speed evaluation (Section Fan Speed and Airflow Rate).

#### 2.2.5. Statistical Analysis

Analysis of variance (ANOVA) and Levene’s test for homogeneity of variance were performed using SPSS version 10.0 for Windows (SPSS Inc., Chicago, IL, USA). Post-hoc testing (*p* < 0.05) of the multiple comparisons was performed by either the Scheffé or Games–Howell test depending on whether Levene’s test was insignificant or significant, respectively.

## 3. Results and Discussion

### 3.1. DPI Add-On Device Prepared by 3D Printing

The DPI add-on equipment was printed using ABS as the printing material and it was readily assembled with the Accuhaler as presented in Figure 4a. The printed product had a smooth surface, and each printed part could be firmly joined together as specified. The add-on device was pre-tested to ensure that it worked properly. Drug particles were observed to be expelled from the assembled DPI after pressing the yellow switch on the add-on equipment, as shown in Figure 4b,c.

### 3.2. HPLC Assay Method Validation

HPLC assay to simultaneously quantify salmeterol xinafoate and fluticasone propionate was validated by establishing the method’s specificity, range, linearity, precision, and accuracy.

#### 3.2.1. Specificity

The retention times of salmeterol xinafoate, fluticasone propionate, and the standard reference mixture solution were analyzed to evaluate the specificity of the HPLC assay. First, λmax of salmeterol xinafoate and fluticasone propionate were scanned by the HPLC-PDA detector. The λmax for salmeterol xinafoate and fluticasone propionate were 245 nm and 235 nm, respectively. As illustrated in Figure 5a,b, the average retention times of separate injections of salmeterol xinafoate and fluticasone propionate were 3.927 min and 4.80 min, respectively. The solution mixture containing both drugs were assayed at 245 nm for salmeterol xinafoate and 235 nm for fluticasone propionate, and the retention times of the two drugs were found to be comparable to the retention times of the separately analyzed pure drugs (Figure 5c). The HPLC chromatogram of the mixture solution showed complete peak resolution suggesting satisfactory peak separation of the two drugs. This is in agreement with a previous report [23] and demonstrates the specificity of the HPLC assay.

#### 3.2.2. Linearity and Range

Calibration graphs with respective correlation coefficients (R^2^) of 0.9999 and 1.0000 were obtained using 5 standard solutions of salmeterol xinafoate (0.1875–1.125 µg/mL) and 5 standard solutions of fluticasone propionate (0.625–3.750 µg/mL), and these are presented in Figure 6a,b, respectively. The HPLC assay for the two drugs was deemed to exhibit linearity over the respective drug concentration range.

#### 3.2.3. Accuracy

The accuracy of the HPLC assay was evaluated by analyzing in triplicates samples (combined solution containing two model drugs) that had been spiked with five different concentrations of salmeterol xinafoate (0.1875–1.125 µg/mL) and fluticasone propionate (0.625–3.750 µg/mL). Drug concentrations in each sample were determined using the calibration curve, and the percent recovery was calculated (Table 3). The percent recovery was 99.65 to 106.93% for the spiked salmeterol xinafoate samples and 99.57 to 101.16% for the spiked fluticasone propionate samples. On this basis, as the percent recovery of all spiked samples was within the acceptable AOAC 2019 range of 80 to 110% [22]. It can be concluded that the analytical method was accurate.

#### 3.2.4. Precision

The precision of the HPLC assay was investigated by measuring the AUC of the drug peaks obtained for a standard solution that was consecutively analyzed six times in the HPLC (Table 4). RSD for the salmeterol xinafoate and fluticasone propionate peak areas were 0.3734% and 0.1486%, respectively. The acquired percent RSD was less than 7.30%, suggesting that the assay has met with the precision defined in accordance with the 2019 AOAC standard [22].

### 3.3. Fan Speed and Airflow Rate

The relationship between the motor power, fan speed, and airflow of the add-on device was analyzed since these characteristics have a direct impact on the particle flow from the DPI device. The percent motor power was adjusted using the rotation angle of the PWM speed controller and the effects of the percent motor power on the fan speed and flow rate of the add-on device are presented in Figure 7. The fan speed and airflow rate were increased when the percent power was raised. At low percent power of up to 50%, there was a near-linear relationship between fan speed and percent power, and between airflow and percent power. However, at high power of between 80 and 100%, the effects of motor power on the fan speed and airflow were significantly muted, and this might be due to the generation of air turbulence within the device that consequently affected the fan speed and airflow [24]. The maximum fan speed and airflow rate of the add-on equipment were, respectively, 20,549.33 ± 104.74 rpm and 12.73 ± 0.06 L/min, and these were achieved at 180-degree rotation angle of the PWM speed controller (equivalent to 100% power).

### 3.4. Emitted Dose Uniformity

The amounts of drugs (salmeterol xinafoate and fluticasone propionate) emitted from the DPI alone, and from the DPI attached with the add-on equipment were captured using the USP Apparatus B sampling apparatus and quantified using the validated HPLC assay. The emitted salmeterol xinafoate and fluticasone propionate doses from the DPI with the add-on device with different fan power are shown in Figure 8a,b, respectively. It could be demonstrated that the emitted drug dosages were dependent on the pump flow (simulating inhalation flow rate) and fan speed. High pump flow (60 L/min) increased both the salmeterol and fluticasone dosages emitted. Reducing the flow rate of the pump from 60 to 30 or 10 L/min lowered the dosages emitted by 10–15 mg for salmeterol and 50 mg for fluticasone. This implies that patients with limited inhaling capability (inhalation flow rate less than 60 L/min) may not be able to access the amount of medicine required, and have the potential to experience reduced therapeutic efficacy. Attaching the DPI to the add-on equipment enabled the utilization of airflow generated in the add-on equipment to improve the emitted drug doses. In agreement with Figure 7, the 50% motor power provided optimum drug release as the high fan speed generated at 100% power might cause air flow turbulence. At 50% fan motor power, the released salmeterol significantly increased % (*p*-value < 0.01) to 5–30 mg (10–60%,), while the emitted fluticasone climbed (*p*-value < 0.0001) to 50–70 mg (30–35%) for all pump flow rates (10–60 L/min). The difference in emitted doses of salmeterol xinafoate and fluticasone propionate might be attributed to their particle size. The aerodynamic diameter of fluticasone propionate (3.2 µm) was smaller than that of salmeterol xinafoate (3.5 µm) [25]. This increases the rate at which the add-on device’s pump and fan expel fluticasone propionate particles. At 50% fan power and 30 L/min flow rate, salmeterol xinafoate and fluticasone propionate emitted 70.40 ± 0.74 µg (141.48% of label amount) and 229.18 ± 2.18 µg (91.60% of label amount), respectively. This demonstrates that the new prototype DPI add-on equipment can increase the emitted drug doses for patients with poor inhalation velocities, such as those with COPD and the elderly, who are typical DPI users. Increasing and maintaining the pharmacological dose can improve disease states and treatment results in patients [26].

### 3.5. Air Flow Simulation by Computational Fluid Dynamics

The modification in air flow pattern in the DPI brought on by the attachment of the add-on device was studied using Solidworks flow simulation, a computational fluid dynamics (CDF) program [27]. The computer-aided design (CAD) modelling of the Accuhaler with or without the add-on equipment is shown in Figure 3a,b, respectively.

Figure 9 depicts the flow streamlines during operation. In the regular Accuhaler without the add-on device, air passed from the Accuhaler air inlet to a sealed blister pocket to expel the drug powder. After that, the air and drug powder were emitted through the by-pass channel that is aligned with the mouthpiece (Figure 9a). The Accuhaler with add-on equipment, on the other hand, experienced an initial flow from the fan inlet when the add-on equipment’s fan was turned on. The air was compressed and forced into the air inlet of the Accuhaler. Following that, the airflow in the Accuhaler was the same as that in the Accuhaler without the add-on equipment (Figure 9b).

In the simulation program, the fan speed of the add-on equipment was varied from 50% (15,031 RPM) to 100% (20,549.33 RPM) of motor power to assess the effect of fan rotation on the pressure and air velocity at different positions in the Accuhaler. As presented in Table 5 and Figure 10a, the result reveals that the average pressure drop across the blister pocket was greater in the Accuhaler with add-on equipment than in the Accuhaler without add-on equipment. This pressure drop also tended to increase as the outlet volume flow increased, and the higher pressure drop could aid in powder fluidization, entrainment and particle re-suspension [28]. At the Accuhaler inlet, higher pressures were generated when the add-on equipment was attached than when the add-on equipment was absent, signifying greater air velocity at the blister pocket when the add-on equipment was in use. Table 5 and Figure 10b demonstrate that when this auxiliary equipment was utilized, the output volume flow and average air velocity of the circulation through the blister pocket increased. Higher air velocity suggests that a larger amount of force was applied to the drug particles, resulting in a more effective dosage emission [28]. Furthermore, differing inhalation rates (outlet volume flow) can cause large pressure decreases and air velocity differences at the blister pocket. The pressure drop and air velocity are significantly lower at 10 and 30 L/min inhalation flow rates than at 60 L/min. The DPI add-on device, on the other hand, could increase pressure drop and air velocity at any inhalation rate, implying that it could be utilized for patients with varying inhalation abilities.

The contour plots of the air velocity were simulated using Solidworks flow (Figure 11) corresponding with the airflow velocity result in Table 5. The red hue in the contour plots denotes a high air velocity (20,000 m/s), whereas the blue color denotes a low air flow. Higher air velocity at the mouthpiece region was achieved by increasing the outflow volume flow (inhalation rate). The mouthpiece, blister, and air path before the blister can all benefit from increased fan power. By raising the air velocity, the amount of drug particles that migrate out of the blister pocket can be enhanced [29].

## 4. Conclusions

Based on the results, it is reasonable to infer that the novel DPI add-on equipment created using the 3D printing process could increase the emitted drug dose from a DPI for patients with low inspiratory rates. The add-on equipment’s air flow rate could well be modified to accommodate varying levels of patient inhalation capacity. This study has shown that with the appropriate fan motor adjustment, the amounts of drugs discharged from a DPI attached with the equipment could be increased by 15–20%. This demonstrates the potential for this add-on equipment model to improve the therapeutic efficacy of DPI in COPD patient groups, geriatric populations, and children with a low or inadequate inhalation rate. This add-on device, which is simple to prepare, inexpensive, and easy to assemble with the Accuhaler, has the potential to be manufactured by local industry and can be used in a variety of settings, such as hospitals or pharmacies, where doctors or pharmacists could diagnose or offer advice to groups of patients who have difficulty inhaling DPI drugs on their own. However, the in vivo clinical study is necessary for the following phase in order to establish that this innovative add-on device application is indeed feasible.

## Figures and Tables

**Figure 1 pharmaceutics-14-01922-f001:**
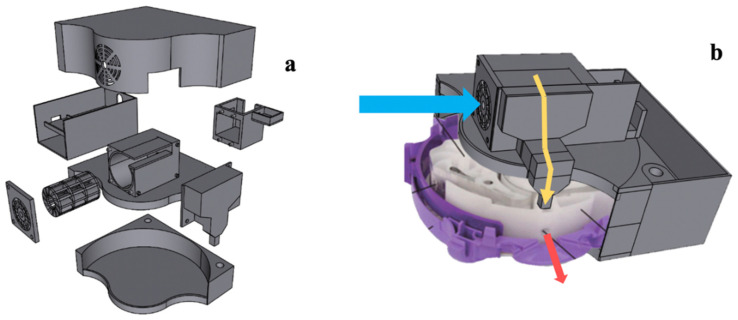
(**a**) The DPI add-on device model design and (**b**) the assemble of DPI and the add-on device.

**Figure 2 pharmaceutics-14-01922-f002:**
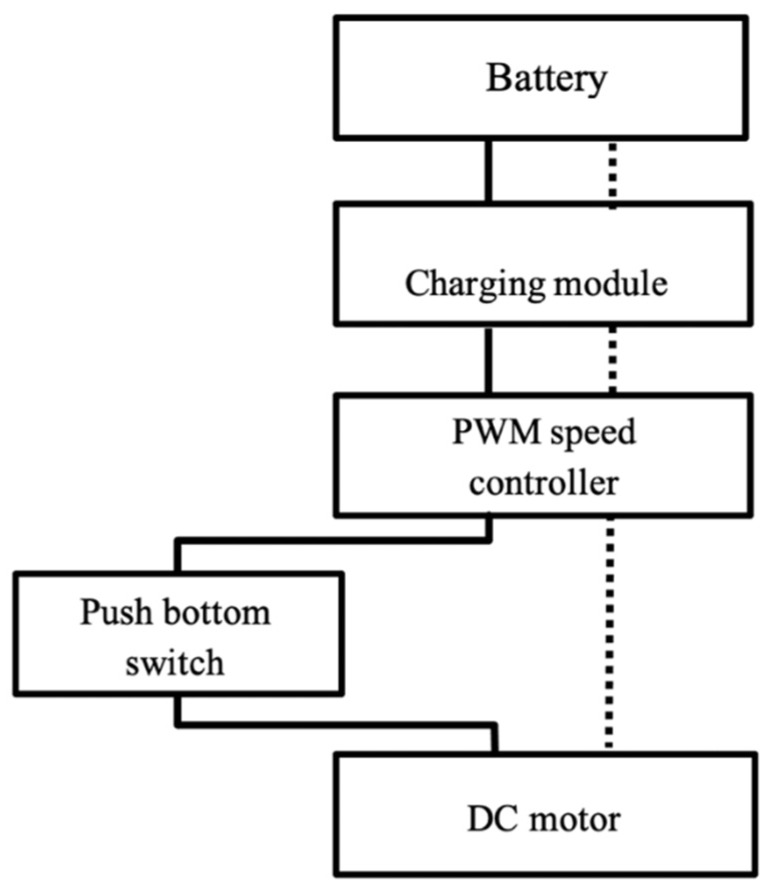
DPI add-on device electrical circuit.

**Figure 3 pharmaceutics-14-01922-f003:**
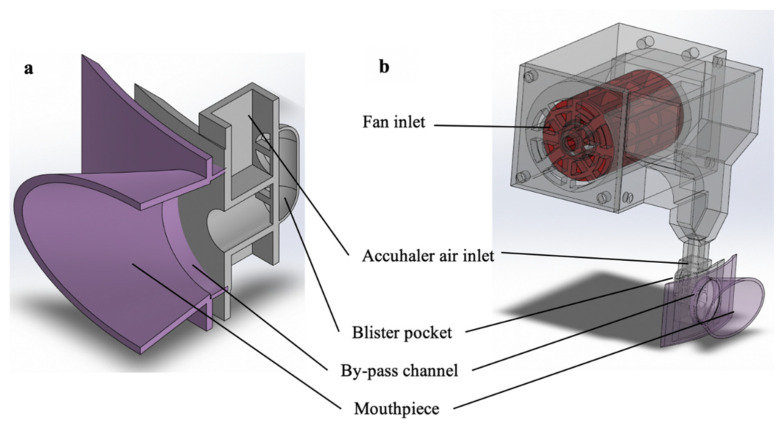
CAD modelling of (**a**) the Accuhaler and (**b**) Accuhaler with add-on device.

**Figure 4 pharmaceutics-14-01922-f004:**
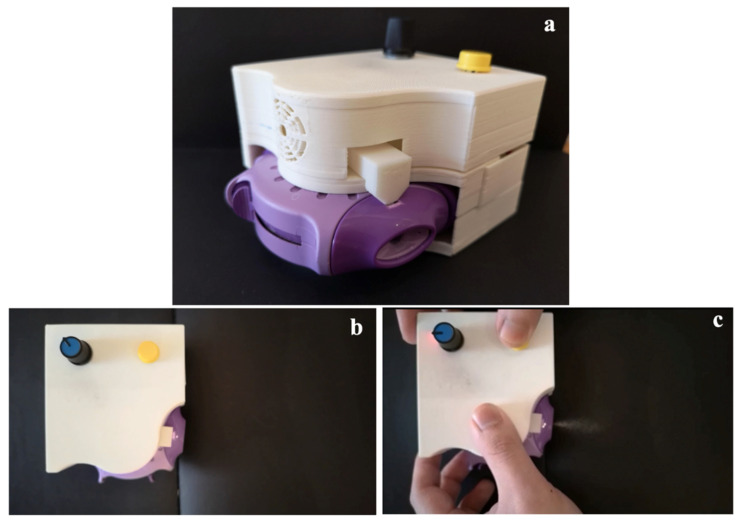
(**a**) Assembly of the 3D printed add-on device and Accuhaler and (**b**,**c**) expulsion of drug from the Accuhaler upon actuation of the yellow button on the add-on equipment.

**Figure 5 pharmaceutics-14-01922-f005:**
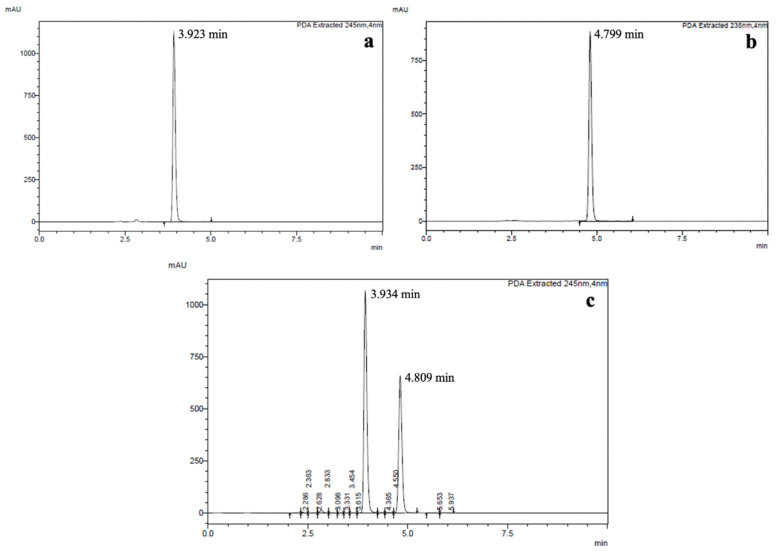
HPLC chromatogram of (**a**) salmeterol xinafoate, (**b**) fluticasone propionate, and (**c**) their mixture.

**Figure 6 pharmaceutics-14-01922-f006:**
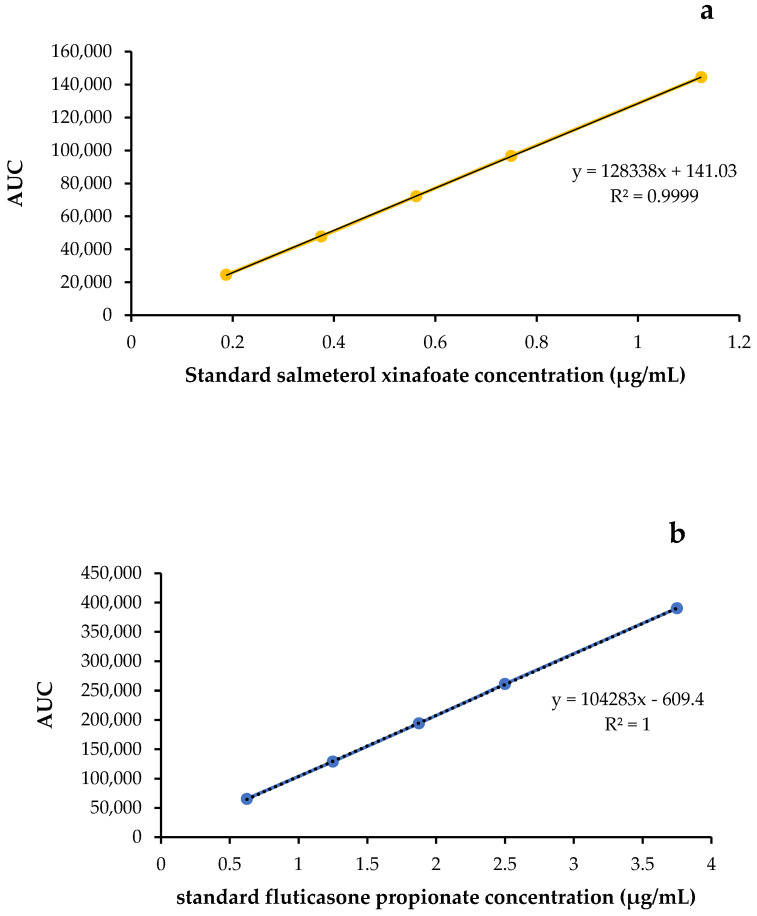
HPLC calibration curves of (**a**) salmeterol xinafoate and (**b**) fluticasone propionate.

**Figure 7 pharmaceutics-14-01922-f007:**
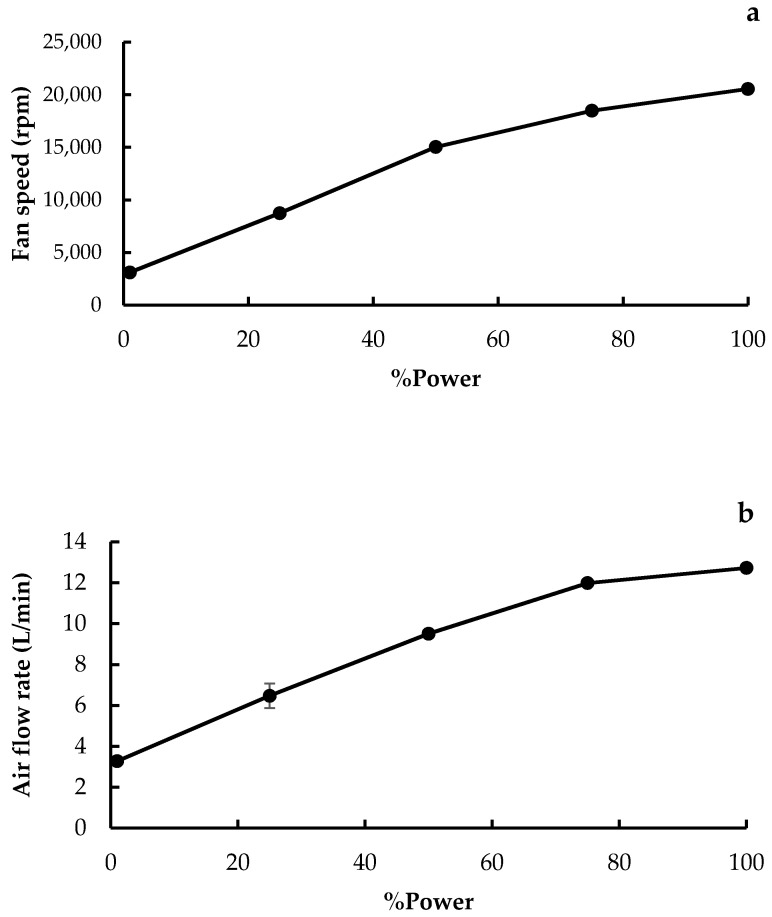
Relation of %power to (**a**) fan speed and (**b**) airflow rate of the add-on device.

**Figure 8 pharmaceutics-14-01922-f008:**
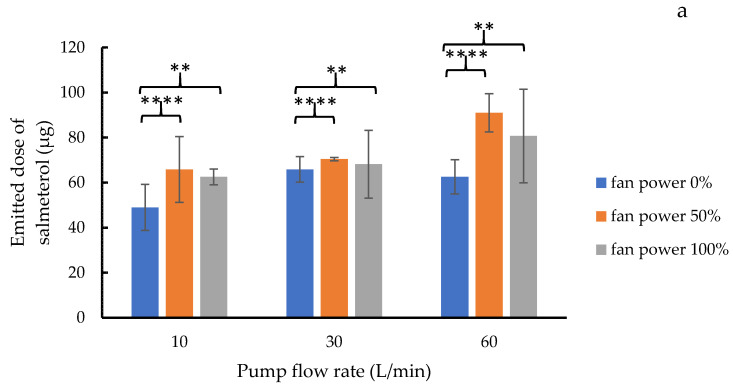
Relation between the fan motor power and emitted dose of (**a**) salmeterol xinafoate and (**b**) fluticasone propionate from a DPI attached with the add-on equipment. Statistical analysis was performed using an ordinary two-way ANOVA and comparing 0% fan power only at each pump flow rate with 50–100% fan power for emitted dose uniformity. (** *p*-value < 0.01, **** *p*-value < 0.0001).

**Figure 9 pharmaceutics-14-01922-f009:**
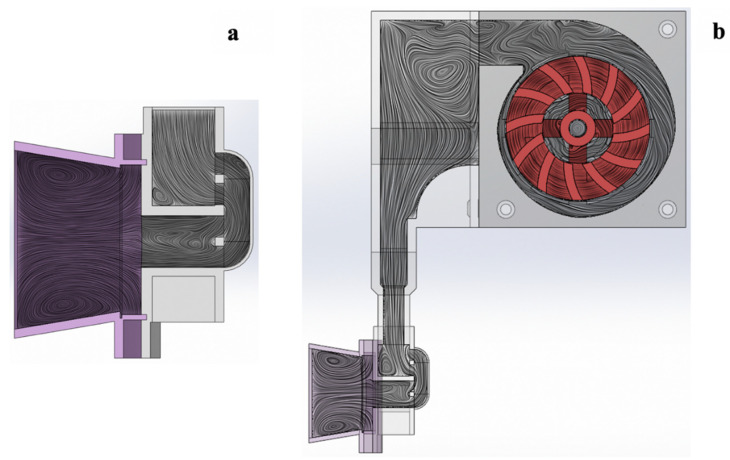
Airflow streamlines in the (**a**) Accuhaler alone, and (**b**) Accuhaler attached with add-on equipment as determined by CFD simulation.

**Figure 10 pharmaceutics-14-01922-f010:**
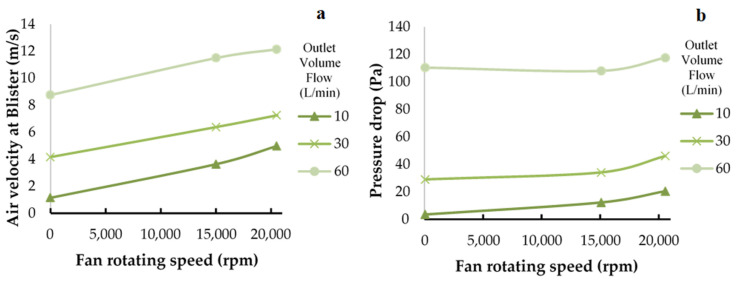
The relationships between fan rotating speed, (**a**) airflow velocity and (**b**) pressure drop at the blister pocket for Accuhaler with or without add-on device were determined through CFD simulation.

**Figure 11 pharmaceutics-14-01922-f011:**
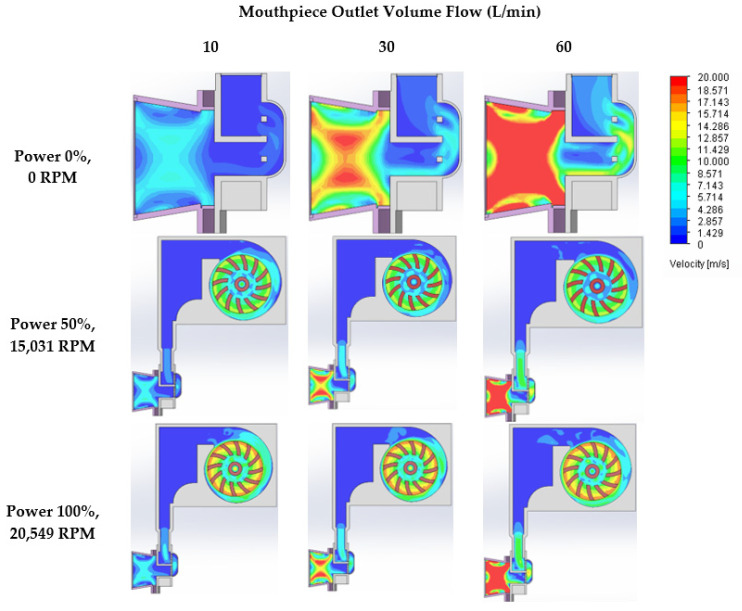
The contour plots of the air velocity with different mouthpiece outlet volume flow and fan rotating speed as determined through CFD simulation.

**Table 1 pharmaceutics-14-01922-t001:** 3D-printing parameters.

Parameter	Condition
Model material	ABS (acrylonitrile–butadiene–styrene)
Nozzle temperature	230–240 °C
Heated Bed temperature	90–100 °C
Printing tpeed	30–50 mm/s
Cooling Fan	Off

**Table 2 pharmaceutics-14-01922-t002:** HPLC operational conditions for the assay of salmeterol xinafoate and fluticasone propionate in a sample.

Parameters	Condition
Column	Particle size 5 µm, pore size 100 Ǻ 150 × 4.6 mm, C18 (ACE, Reading, UK)
Mobile phase	Acetonitrile and buffer solution (70:30)
Buffer solution	1.0 g potassium dihydrogen phosphate in 1000 mL deionized water adjusted to pH 3 with ortho-phosphoric acid.
Flow rate	1 mL/min
Detector	PDA detector
Injection volume	40 µL
Column temperature	40 °C
Run time	10 min

**Table 3 pharmaceutics-14-01922-t003:** Theoretical spiked concentration, the average actual determined concentration, and the percent recovery of salmeterol xinafoate and fluticasone propionate.

Reference Standard	Theoretical Concentration Spiked (µg/mL)	Average Actual Determined Concentration (µg/mL), *n* = 3	%Recovery
Salmeterol xinafoate	0.1931	0.1940	100.4648
	0.3787	0.3796	100.2216
	0.5644	0.5646	100.0399
	0.7500	0.7474	99.6514
	1.1213	1.1990	106.9298
Fluticasone propionate	0.6436	0.6461	100.3947
	1.2624	1.2644	100.1613
	1.8812	1.8750	99.6706
	2.5000	2.4893	99.5715
	3.7376	3.7810	101.1605

**Table 4 pharmaceutics-14-01922-t004:** AUC (mean ± SD, *n* = 6) and %RSD for salmeterol xinafoate and fluticasone propionate determined from the HPLC assay.

Drugs	AUC (Mean ± SD, *n* = 6)	%RSD
Salmeterol xinafoate	96875 ± 361.75	0.3734
Fluticasone propionate	260963.67 ± 387.81	0.1486

**Table 5 pharmaceutics-14-01922-t005:** Pressure profile and Airflow velocity at the blister pocket in Accuhaler with or without add-on device as determined through CFD simulation.

Inhalation Flow Rate (L/min)	Fan Rotation (rpm)	Pressure Average at Accuhaler Inlet (Pa)	Pressure Average at Blister (Pa)	Pressure Drops (Pa)	Airflow Velocity Average at Blister (m/s)	Number of Iterations
10	0	101,324.86	101,321.34	3.52	1.146	82
	15,031	101,343.54	101,331.29	12.25	3.638	305
	20,549.33	101,361.86	101,341.37	20.49	4.984	279
30	0	101,323.68	101,294.74	28.94	4.162	84
	15,031	101,331.00	101,296.95	34.05	6.387	309
	20,549.33	101,356.66	101,310.62	46.04	7.261	288
60	0	101,319.85	101,209.52	110.33	8.765	83
	15,031	101,290.49	101,182.56	107.93	11.504	476
	20,549.33	101,313.14	101,195.66	117.48	12.145	391

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
