# Peer review of "Development of an Add-On Device Using 3D Printing for the Enhancement of Drug Administration Efficiency of Dry Powder Inhalers (Accuhaler)"

_pharmaceutics, 2022, doi:10.3390/pharmaceutics14091922_

Round 1

Reviewer 1 Report

This is a very interesting and well-written manuscript regarding the improvement of drug delivery via an 3DP add-on device for inhalers.

The authors should address the following issues:

1.      Line 19: Replace “to develop via 3-D printing of an add-on device for dry powder 19 inhalers (Accuhaler)” with “to develop an add-on device for dry powder inhalers (Accuhaler) via 3D printing”

2.      Line 24: “an HPLC assay”

3.      Line 74: Replace “was designed with” with “was designed containing”

4.      Line 75: Delete the word “was”.

5.      Line 286: Replace word “more” with “significantly”

6.      Line 287: Replace word “then” with “consequently”

7.      Line 314: Replace phrase “because of” with “be attributed to”

8.      It would be desirable to specify potential applications in various settings (clinical, community, industry), regarding the feasibility of implementing this add-on device in each setting.

Author Response

-Reviewer 1

This is a very interesting and well-written manuscript regarding the improvement of drug delivery via an 3DP add-on device for inhalers.

Response: The authors would like to thank the reviewer for the positive feedback. We hereby acknowledge all of the comments made by the reviewer and have provided a point-to-point response to all his/her comments with a yellow highlight.

The authors should address the following issues:

  1. Line 19: Replace “to develop via 3-D printing of an add-on device for dry powder 19 inhalers (Accuhaler)” with “to develop an add-on device for dry powder inhalers (Accuhaler) via 3D printing”

Response: The sentence has been corrected.

  1. Line 24: “an HPLC assay”

Response: The sentence has been corrected.

  1. Line 74: Replace “was designed with” with “was designed containing”

Response: The sentence has been corrected.

  1. Line 75: Delete the word “was”.

Response: The sentence has been corrected.

  1. Line 286: Replace word “more” with “significantly”

Response: The sentence has been corrected.

  1. Line 287: Replace word “then” with “consequently”

Response: The sentence has been corrected.

  1. Line 314: Replace phrase “because of” with “be attributed to”

Response: The sentence has been corrected.

  1. It would be desirable to specify potential applications in various settings (clinical, community, industry), regarding the feasibility of implementing this add-on device in each setting.

Response: The potential applications of the developed add-on device have been included to the conclusion part.

Reviewer 2 Report

Comments pharmaceutics-1885643

In this manuscript, the authors employed FDM 3D printing to prepare an add-on device of dry powder inhalers. The idea is interesting, and the manuscript can be considered for publication after addressing the following comments:

1.     English of the manuscript needs to be improved; numerous grammatical errors can be found.

2.     A brief introduction of 3D printing is recommended to be included to explain why choosing 3D printing instead of other technologies.

3.     For the Emitted Dose Uniformity, how many replicates have been performed?

4.     Were any statistics analysis performed? Because no statistical results can be found in the results section.

5.     Figure 9a, the standard deviation of each pump flow rate seems to be very large, is it statistically significant to say that it is different from 0% or 50% power?

6.     Table 5 can authors include standard deviation?

Author Response

-Reviewer 2

In this manuscript, the authors employed FDM 3D printing to prepare an add-on device of dry powder inhalers. The idea is interesting, and the manuscript can be considered for publication after addressing the following comments:

Response: The authors would like to thank the reviewer for the positive feedback. We hereby acknowledge all of the comments made by the reviewer and have provided a point-to-point response to all his/her comments with a yellow highlight.

  1. English of the manuscript needs to be improved; numerous grammatical errors can be found.

Response: The grammatical errors have been corrected in the manuscript.

  1. A brief introduction of 3D printing is recommended to be included to explain why choosing 3D printing instead of other technologies.

Response: The review of 3D printing and the reason for selecting 3D printing instead of other technologies have been addressed in the introduction part.

  1. For the Emitted Dose Uniformity, how many replicates have been performed?

Response: For the Emitted Dose Uniformity evaluation, the experiment was conducted three times for each test run. Information regarding replication has been included in the method section of the manuscript.

  1. Were any statistics analysis performed? Because no statistical results can be found in the results section.

Response: Statistical analysis was performed using ordinary two-way ANOVA and comparing 0% fan power only at each pump flow rate with 50%–100% fan power for emitted dose uniformity.

  1. Figure 9a, the standard deviation of each pump flow rate seems to be very large, is it statistically significant to say that it is different from 0% or 50% power?

Response: Figure 9 was revised, and the statistical analysis was included in the figure and its caption. At 50% fan motor power, the released salmeterol significantly increased (from 0% fan motor power) to 5-30 mg (10-60%, P-value<0.01), while the emitted fluticasone climbed to 50-70 mg (30-35%, P-value<0.0001) for all pump flow rates (10-60 L/min).

  1. Table 5 can authors include standard deviation?

Response: Table 5 contains data from Computational Fluid Dynamics (CFD) program which is not actual experiment. CFD is a numerical method for data analysis (using mathematical equation); therefore, it is not reported as a standard deviation. Data is calculated repeatedly in CFD until iteration convergence occurs. Iteration means the act of repeating a process to approach the desired goal, target, or result. Therefore, the number of iterations should be reported to indicate reliability. The number of iterations has been added to Table 5.

Reviewer 3 Report

The study developed an add-on device for dry powder inhalers (Accuhaler) via 3-D printing to improve drug administration efficiency in patients with limited inspiratory capacity.

The paper presents an innovative method for effective dosage emission—a technique that may assist children, the elderly, and people with COPD.

The researchers guide the reader through the research process and visually explain every step. They stress the device’s merits and efficiencyaccommodating varying levels of patient inhalation capacity. However, I advise also emphasizing the study’s limitations.

Furthermore, I recommend adding academic background by including papers about the 3D printing of pharmaceuticals or 3D printing technology to improve drugs’ quality.

Please delete the following paragraph on page 7, section 3: “This section may be divided by subheadings. It should provide a concise and precise 210 description of the experimental results, their interpretation, as well as the experimental 211 conclusions that can be drawn.”

Author Response

-Reviewer 3

The study developed an add-on device for dry powder inhalers (Accuhaler) via 3-D printing to improve drug administration efficiency in patients with limited inspiratory capacity.

The paper presents an innovative method for effective dosage emission—a technique that may assist children, the elderly, and people with COPD.

The researchers guide the reader through the research process and visually explain every step. They stress the device’s merits and efficiency—accommodating varying levels of patient inhalation capacity. However, I advise also emphasizing the study’s limitations.

Response: The study’s limitations has been addressed in the conclusion part.

Furthermore, I recommend adding academic background by including papers about the 3D printing of pharmaceuticals or 3D printing technology to improve drugs’ quality.

Response: The review of 3D printing and the reason for selecting 3D printing instead of other technologies have been addressed in the introduction part.

Please delete the following paragraph on page 7, section 3: “This section may be divided by subheadings. It should provide a concise and precise 210 description of the experimental results, their interpretation, as well as the experimental 211 conclusions that can be drawn.”

Response: The incorrect paragraph was eliminated.

Reviewer 4 Report

In the present manuscript, authors have explored " Development of an Add-On Device using 3D Printing for the Enhancement of Drug Administration Efficiency of Dry Powder Inhalers (Accuhaler)". The subject is of interest and falls in the topics of Parmaceutics Journal. The study lacks a specific hypothesis being tested. There is poor focus with too many things going on.

After reviewing the manuscript thoroughly, I have following comments:

Absract: Full form of DPI is missing

Figure 3 is not clear.

Section 2.2.3.3. Why Photo Diode detector is employed in the HPLC assay.

Table 2: Is the pH 3 suitable for column? Is the column is safe at this pH?

Define Emitted Dose Uniformity and its significant.

 What is the significance of the use of both APIs (salmeterol xinafoate and fluticasone propionate). Their comparison is missing in the manuscript. The proportion of the both moieties employed is also is not presented.

How Add-On Device using 3D Printing was developed for dry powder inhalers?

What is mechanism behind the improvement in the drug administration efficiency of these moieties?

Author Response

-Reviewer 4

In the present manuscript, authors have explored " Development of an Add-On Device using 3D Printing for the Enhancement of Drug Administration Efficiency of Dry Powder Inhalers (Accuhaler)". The subject is of interest and falls in the topics of Parmaceutics Journal. The study lacks a specific hypothesis being tested. There is poor focus with too many things going on.

Response: The authors would like to thank the reviewer for the valuable feedback. We hereby acknowledge all the comments made by the reviewer and have provided a point-to-point response to all of his/her comments with a yellow highlight.

After reviewing the manuscript thoroughly, I have following comments:

  1. Absract: Full form of DPI is missing

Response: The full form of DPI has been added to the abstract.

  1. Figure 3 is not clear.

Response: Figure 3 was revised for better image resolution. 

  1. Section 2.2.3.3. Why Photo Diode detector is employed in the HPLC assay.

Response: The photo diode array (PDA), also known as the diode array detector (DAD) can measure the entire wavelength range in real time. In this study, Photodiode Array Detector was used to quantified salmeterol xinafoate and fluticasone propionate. The assay method was adapted from USP38 (Fluticasone Propionate and Salmeterol Inhalation Powder monograph). Therefore, the assay method of salmeterol xinafoate and fluticasone propionate were validated, and the results were acceptable according to the standard method performance requirements-AOAC International methods committee guidelines.

  1. Table 2: Is the pH 3 suitable for column? Is the column is safe at this pH?

Response: The HPLC assay method was applied from USP 38 and the previous work by Serkan Acar and colleagues. To confirm the suitability and correctness of the HPLC Assay method, The method was validated on the topics of specificity, range & linearity, accuracy, and precision. The result found that this method is acceptable under the AOAC 2019 standard.

  1. Define Emitted Dose Uniformity and its significant.

Response: The Emitted Dose Uniformity is a method for evaluating the uniformity of each administered dose of medication. This is a critical evaluation subject since it determines the amount of medication a patient can take every inhalation. Because the developed add-on equipment affects the flow of medication particles, it is essential to certify that all or the vast majority of drugs were delivered. The definition of Emitted Dose Uniformity and its significant were addressed in the manuscript.

  1. What is the significance of the use of both APIs (salmeterol xinafoate and fluticasone propionate). Their comparison is missing in the manuscript. The proportion of the both moieties employed is also is not presented.

Response: Salmeterol xinafoate and fluticasone propionate are the active drug in the Accuhaler product. Therefore, it is important to monitor both APIs to evaluate effect of inhale efficiency and the fan power of the add-on equipment. The comparison discussion between of the emitted drugs (salmeterol xinafoate and fluticasone propionate) has been added to

Section 3.4, Emitted Dose Uniformity.

  1. How Add-On Device using 3D Printing was developed for dry powder inhalers?

Response: The DPI add-on device was developed to help patients (small children, the elderly, and those with severe asthma and COPD) who lack adequate inhalation efficiency to administer DPI. The interior and external dimensions, shape, and operating mechanism of the Accuhaler were initially investigated by pharmaceutical company. The accessory was then developed using the Shapr3D application. The add-on device was produced using Fused Deposition Modeling (FDM) 3D printing since it is user-friendly, inexpensive, and printing materials are readily available.

  1. What is mechanism behind the improvement in the drug administration efficiency of these moieties?

Response: According to the airflow streamlines created by CFD simulation, the airflow was generated from fan in the add-on device. Then, the generated air flowed through the drug pocket (Figure 10 a) and carried the drug out.

Round 2

Reviewer 2 Report

The manuscript has been improved and can be considered accepted after amending the following comments:

1. Lines 70-71, please correct it to "Nowadays, preparing conceptual add-on devices can be possible by using 3D printing technology"

2. Abbreviation of FDM should be used in line 78 as it first appeared instead of line 81

Reviewer 4 Report

In the present manuscript, the authors have explored " Development of an Add-On Device using 3D Printing for the Enhancement of Drug Administration Efficiency of Dry Powder Inhalers (Accuhaler)". The subject is of interest and falls in the topics of Pharmaceutics Journal. The study is acceptable in its present form with minor English corrections.